# RAB5A expression is a predictive biomarker for trastuzumab emtansine in breast cancer

Olav Engebraaten[1,2,3], Christina Yau[4], Kristian Berg [2,5], Elin Borgen[6], Øystein Garred[6], Maria E. B. Berstad[2], Ane S. V. Fremstedal[2], Angela DeMichele[7], Laura van 't Veer [4], Laura Esserman [4] & Anette Weyergang [2✉]

HER2 is a predictive biomarker for HER2-targeted therapeutics. For antibody–drug conjugates (ADCs; e.g., trastuzumab emtansine (T-DM1)), HER2 is utilized as a transport gate for cytotoxic agents into the cell. ADC biomarkers may therefore be more complex, also reflecting the intracellular drug transport. Here we report on a positive correlation between the early endosome marker RAB5A and T-DM1 sensitivity in five HER2-positive cell lines. Correlation between RAB5A expression and T-DM1 sensitivity is confirmed in breast cancer patients treated with trastuzumab emtansine/pertuzumab in the I-SPY2 trial (NCT01042379), but not in the trastuzumab/paclitaxel control arm. The clinical correlation is further verified in patients from the KAMILLA trial (NCT01702571). In conclusion, our results suggest RAB5A as a predictive biomarker for T-DM1 response and outline proteins involved in endocytic trafficking as predictive biomarkers for ADCs.

[1] Department of Oncology, Oslo University Hospital, Oslo, Norway. [2] Institute for Cancer Research, Oslo University Hospital, Oslo, Norway. [3] Institute of Clinical Medicine, University of Oslo, Oslo, Norway. [4] Departments of Surgery and Laboratory Medicine, University of California San Francisco, San Francisco, CA, USA. [5] Department of Pharmacy, University of Oslo, Oslo, Norway. [6] Department of Pathology, Oslo University Hospital, Oslo, Norway. [7] Department of Medicine, Perelman School of Medicine, University of Pennsylvania, Philadelphia, PA, USA. ✉email: anette.weyergang@rr-research.no

The increased focus on personalized medicine has together with our increasing knowledge in cancer biology revealed a great potential for the use of biomarkers in cancer treatment. A number of different biomarkers are already incorporated in clinical practice to predict patient survival, select an appropriate therapy, or monitor disease progression[1]. Predictive biomarkers enable careful selection of those patients most likely to benefit from a specific treatment, and hence, such knowledge is crucial in order to rationally exploit current and future high-cost targeted cancer therapeutics. HER2 (ERBB2) is a validated biomarker in breast cancer and HER2 gene amplification or protein overexpression is found in ~20% of newly diagnosed breast cancer patients[2,3]. HER2 is utilized as a predictive biomarker for treatment with HER2- targeting monoclonal antibodies (mAbs) (trastuzumab and pertuzumab) and tyrosine kinase inhibitors (TKIs) (lapatinib and afatinib)[3]. The pharmacological effects of HER2-targeted mAbs and TKIs are a direct consequence of drug–target interaction and include antibody-dependent cellular cytotoxicity (ADCC) (mAbs), HER2 downregulation, and inhibition of growth-promoting signaling[4–6]. The ability of HER2 to undergo receptor-mediated endocytosis also makes this transmembrane protein a candidate for the delivery of cytotoxic agents into the cancer cells. This has indeed been exemplified by the antibody–drug conjugate (ADC) trastuzumab emtansine (T-DM1)[7,8] which received FDA approval for treatment of metastatic breast cancer in 2013. T-DM1 consists of trastuzumab linked by a thioether (N-maleimidomethyl cyclohexane-1-carboxylate (MCC)) to the highly cytotoxic maytansine-derived drug, DM1[9]. Upon administration, T-DM1 binds to HER2 and is taken into the cell by HER2-mediated endocytosis. Proteolytic degradation of the trastuzumab component within the endo/lysosomal pathway is postulated as the mechanism for cytosolic release of DM1 which subsequently induces microtubule destabilization and cell death[10,11]. T-DM1, therefore, induces a cytotoxic mechanism of action within the cell in addition to the pharmacological effects generated by its trastuzumab component.

The action mechanisms of T-DM1 are clearly more complex than that of HER2-targeting mAbs[12,13] and TKIs and we have evaluated whether this is reflected in the biomarkers that can be used to predict drug response. Candidate biomarkers for T-DM1 efficacy have, until now, focused on HER2 and its downstream signaling in addition to HER3[14,15], and little is known about the impact of proteins involved in endocytosis, endocytic vesicle transport, and exocytosis.

In this work, we report on a correlation between T-DM1 treatment response and RAB5A expression level. Our results show a significant correlation between RAB5A expression and T-DM1 sensitivity in a cell line panel in vitro. These results are further confirmed in T-DM1 treated patients from two independent clinical trials. The present study suggests RAB5A as a predictive biomarker for T-DM1 response.

## Results

**The cellular efficacy of T-DM1 does not correlate to trastuzumab sensitivity.** The antiproliferative effects of the HER2-targeted mAb trastuzumab and the intracellular acting HER2-targeted ADC T-DM1 were established in five HER2-positive cell lines. Subjecting the cells to a 72 h treatment with trastuzumab or T-DM1 revealed the SK-BR-3 and AU-565 cells as highly sensitive to both therapeutics, whereas the SKOV-3 cells, on the contrary, were found non-responsive to trastuzumab and exhibited low sensitivity to T-DM1 as demonstrated by a relatively high IC$_{50}$ of 1.2 μg/ml (Fig. 1 and Table 1). The HCC1954 and MDA-MB-453 cells were both found low- to moderately sensitive to trastuzumab treatment, but responded differently to T-DM1 with

the HCC1954 cells showing high sensitivity, whereas the MDA-MB-453 cells showed low sensitivity (Fig. 1 and Table 1). No clear connection was therefore found between trastuzumab and T-DM1 sensitivity among the five cell lines (Fig. 1 and Table 1).

**The T-DM1 sensitivity correlates to HER2 expression in HER2-positive breast and ovarian cancer cell lines.** Strong HER2 expression was documented in the five HER2-expressing cell lines used in the present study compared to the low expression in MDA-MB-231 reported as HER2 negative (Fig. 2A)[16].

HER2 expression is essential for T-DM1 toxicity. However, it was hypothesized that the level of expression may not necessarily correlate directly to drug sensitivity due to differences in drug processing (e.g., uptake, intracellular transport, and interaction with intracellular drug targets) between the cell lines. Quantification of HER2 expression in the five positive cell lines indicated AU-565 to have the highest expression level of HER2, closely followed by HCC1954 (Fig. 2A, B). A ~50% lower HER2 expression was found in the SK-BR-3 cells compared to AU-565 while SKOV-3 and MDA-MB-453 were identified as the cell lines with the lowest HER2 expression in the panel (0.29 and 0.17 relative expression, respectively) (Fig. 2A, B). The level of HER2 expression reported here is in agreement with other reports[16,17]. Furthermore, a linear relationship was found between HER2 expression and T-DM1 sensitivity among the cell lines, resulting in an R$^2$ value of 0.840 (Fig. 2C).

**HER2-expressing cell lines differ in their expression level of proteins involved in endocytic trafficking.** As T-DM1 is dependent on internalization and intracellular trafficking in order to exert its intracellular mechanism of action, proteins essential for endocytosis and exocytosis were quantified in the cell line panel. These proteins included RAB5A (Fig. 2D, E), implicated in the delivery of cargo from the plasma membrane to early endosomes as well as endosome fusion, RAB4A (Fig. 2D, F), implicated in recycling from early endosomes, and RAB11A (Fig. 2D, G), involved in perinuclear recycling of endosomes and plasma membrane–Golgi traffic[18,19]. The expression level of these proteins among the cell lines showed large differences and no simple connection was found between the expression levels.

**RAB5A protein expression is highly correlated to T-DM1 sensitivity.** It was further assessed whether the investigated proteins involved in endocytosis or exocytosis (Fig. 2D–G) had an impact on T-DM1 sensitivity in the evaluated cell lines. The expression level of RAB5A varied highly between the cell lines (Fig. 2D, E). However, a strong linear correlation was found between T-DM1 toxicity and RAB5A expression among the cell lines (R$^2$ = 0.934) (Fig. 3A). In contrast, no linear correlations were found between T-DM1 sensitivity and expression of RAB4A or RAB11A (Fig. 3B, C). The cellular

**Table 1 Cellular sensitivity of trastuzumab and T-DM1.**

| Cell line | Trastuzumab sensitivity | T-DM1 sensitivity (IC$_{50}$ (μg/ml)) |
|---|---|---|
| SK-BR-3 | +++ | +++ (0.0058) |
| SKOV-3 | − | + (1.2) |
| AU-565 | +++ | +++ (0.0046) |
| HCC1954 | + | +++ (0.0065) |
| MDA-MB-435 | + | ++ (0.12) |

The table shows trastuzumab sensitivity (not sensitive (−), low sensitivity (+), high sensitivity (+++)), and T-DM1 sensitivity (IC$_{50}$ average of three experiments) in the indicated cell lines. Graphic presentations of the viability data are found in Fig. 1. Source data provided.

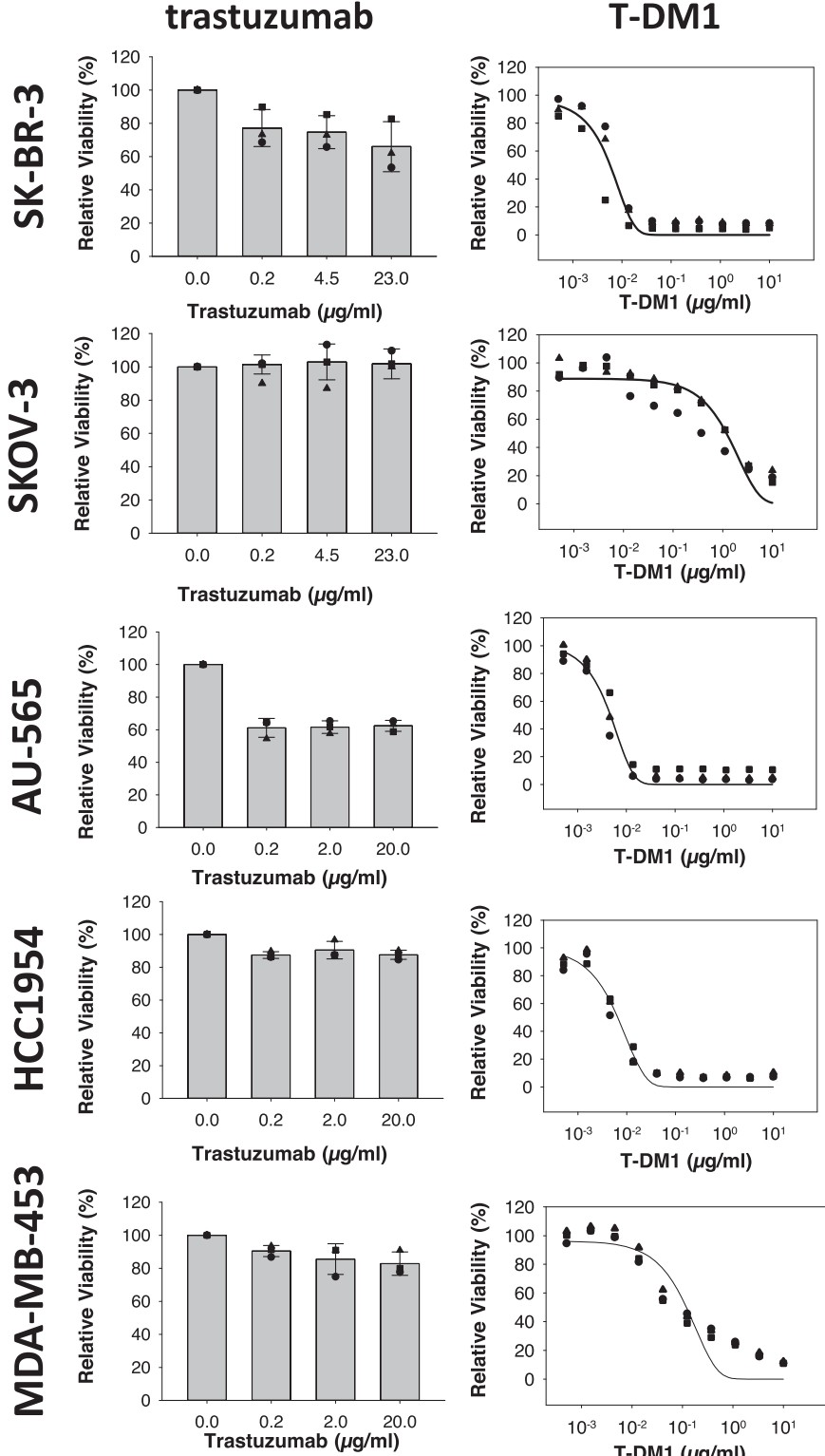

**Fig. 1 In vitro sensitivity to trastuzumab and T-DM1.** Relative viability (MTT) of SK-BR-3, SKOV-3, AU-565, HCC1954, and MDA-MB-453 following 72 h treatments with indicated drugs. The sigmoid curve fit model $a/(1 + \exp(-(x - x_0)/b))$ was used for T-DM1. Data points represent the observed values of three independent experiments, the bar represents the average and the error bars represent the SD(trastuzumab). Source data are provided as a Source Data file.

T-DM1 sensitivity was also correlated to the expression level of HER2 and RAB5A together revealing an $R^2 = 0.962$ (Fig. 3D) which is higher than obtained with the correlations of HER2 ($R^2 = 0.840$) (Fig. 2C) and RAB5A ($R^2 = 0.934$) (Fig. 3A) alone. Thus, T-DM1 sensitivity correlates to the expression of RAB5A in this panel of cell lines, while no linear correlation is found for RAB4A and RAB11A. The expression level of HER2 and RAB5A together serves as a better biomarker for cellular T-DM1 response as compared to either of the two proteins alone.

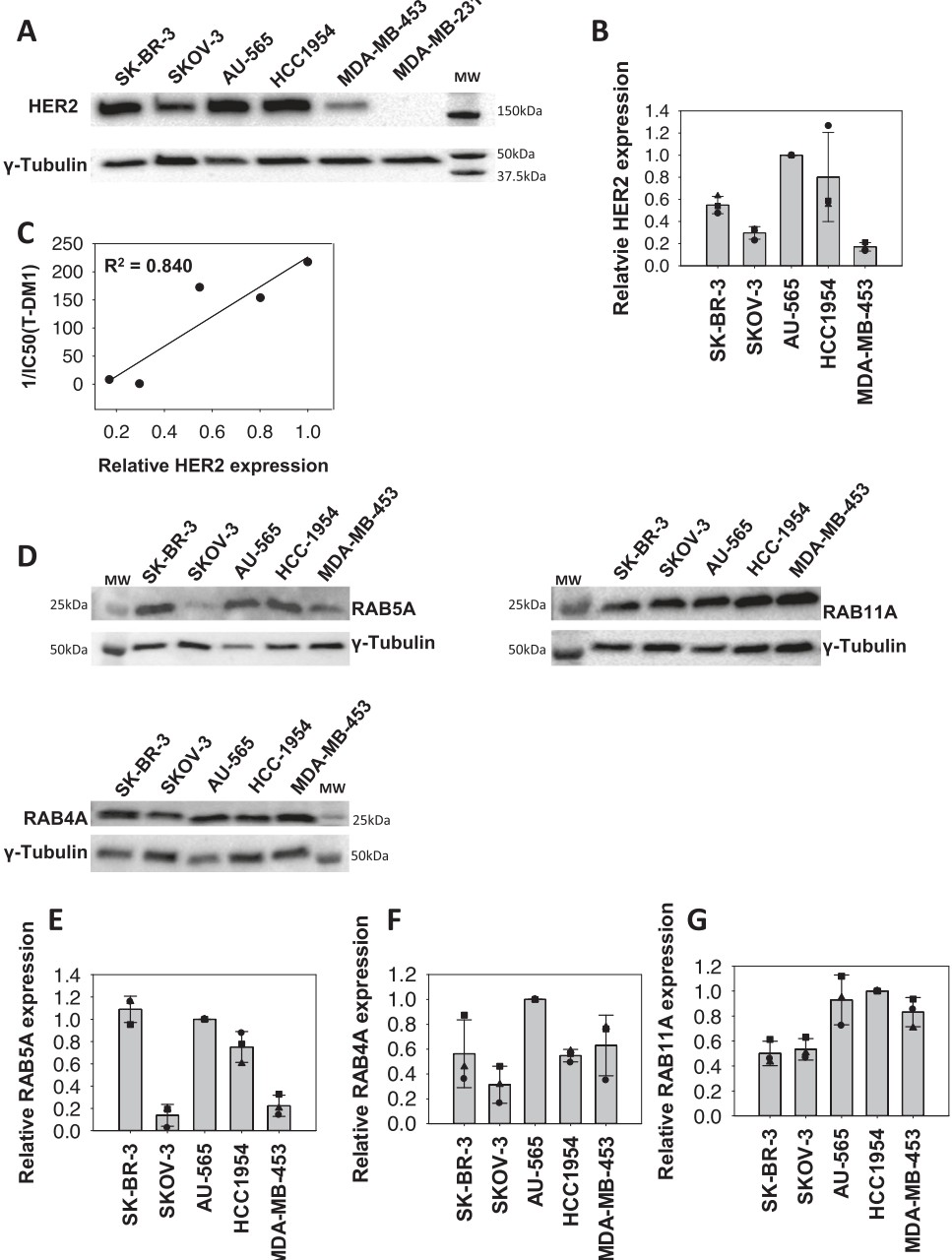

**Fig. 2 HER2 and RAB GTPase expression in cell lines. A** Representative Western blot of HER2 and γ-tubulin expression in SK-BR-3, SKOV-3, HCC1954, AU-565, MDA-MB-453, and MDA-MB-231 cells ($n = 3$). **B** Quantification of the HER2 Western blots relative to those of γ-tubulin. Data points represent the values of three independent experiments, the bars represent the average and the error bars represent the SD of the mean. **C** Linear regression analysis curve of HER2 protein expression and T-DM1 sensitivity ($1/IC_{50}$(T-DM1)). **D** Representative western blot ($n = 3$) of RAB4A, RAB5A, RAB11A, and γ-tubulin expression in SK-BR-3, SKOV-3, AU-565, HCC1954, and MDA-MB-453 cells. **E–G** Quantification of the RAB4, RAB5, and RAB11 Western blots relative to those of γ-tubulin. Data points represent the values of three independent experiments, the bars represent the average and the error bars represent the SD of the mean. Source data are provided as a Source Data file.

**RAB5 RNA expression correlates with T-DM1 sensitivity in the I-SPY2 clinical trial**. We then validated our in vitro finding in the I-SPY2 clinical trial by testing whether the clinical response toward T-DM1, as measured by pCR, could be correlated to RAB5A RNA expression. Figure 4A summarizes the pCR and the hormone receptor (HR) status of the T-DM1 + pertuzumab (T-DM1 + P) and trastuzumab and paclitaxel (TH) control arms. Overall, 30 of the 52 patients on the T-DM1 + P arm and 8 of the 31 patients in the control arm achieved a pCR. Normalized, pre-treatment expression levels of RAB4A, RAB5A, and RAB11A were tested individually as

biomarkers of response. Of the biomarkers evaluated, only RAB5A was associated with response in the T-DM1 + pertuzumab arm (Fig. 4B and Supplementary Table 1, $p = 0.01$, LR test). None of the tested biomarkers were associated with response in the control arm (trastuzumab + paclitaxel) (Fig. 4B). The $p$-value for the interaction between RAB5A expression and treatment was 0.02 and remained <0.05 after adjusting for hormone status (Fig. 4B and Supplementary Table 1).

The association between RAB5A and T-DM1 + pertuzumab response may be attributable to pertuzumab rather than T-DM1.

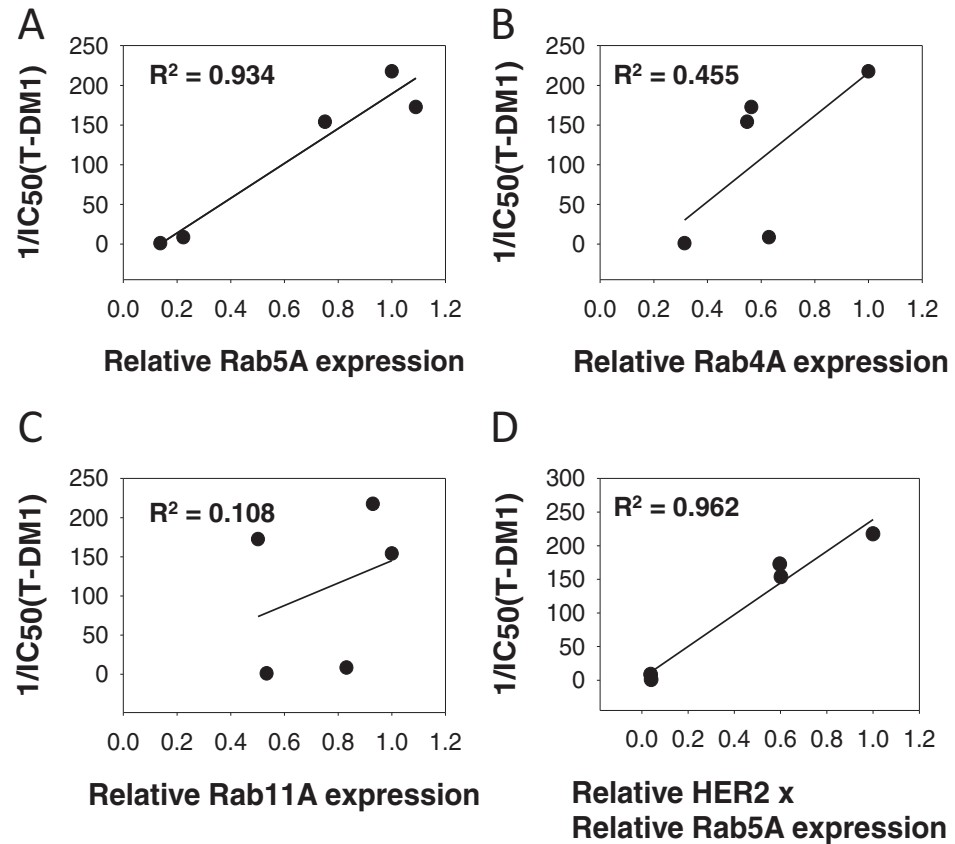

**Fig. 3 T-DM1 and RAB GTPase in vitro correlations.** Linear regression analysis curves between RAB5A (**A**), RAB4A (**B**), and RAB11A (**C**) protein expression (average of $N = 3$ as presented in Fig. 2) and T-DM1 sensitivity ($1/IC_{50}$(T-DM1)) (average of $N = 3$ as presented in Fig. 1 and Table 1) in the five cell lines. **D** The linear regression curve between HER2 (average of $N = 3$ as presented in Fig. 2) × RAB5A protein expression and T-DM1 response. Each data point represents one cell line. Source data are provided as a Source Data file.

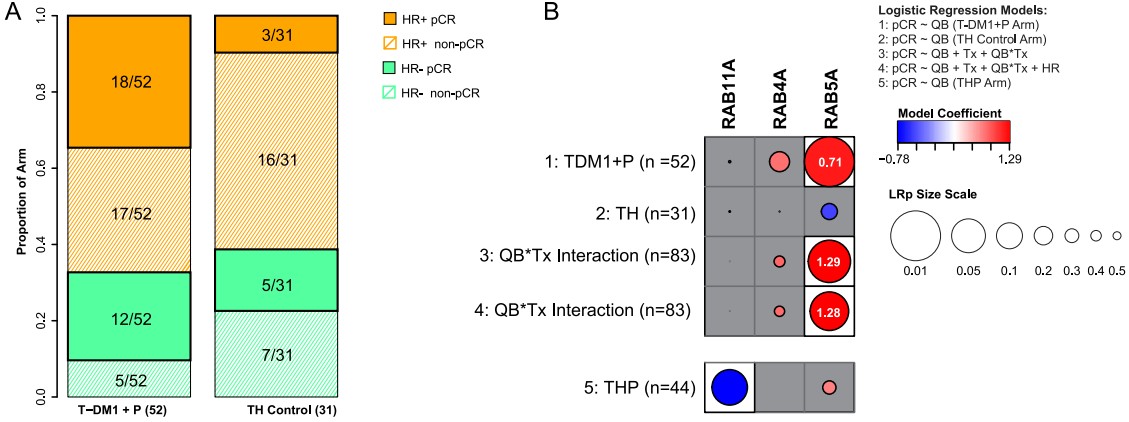

**Fig. 4 T-DM1 and RAB GTPase correlations in the I-SPY2 cohort. A** Number of patients, hormone receptor (HR) status, and pathological complete responses (pCR) in the T-DM1 + pertuzumab and trastuzumab-paclitaxel arm of the I-SPY2 study. **B** Association plot summarizing qualifying biomarker analyses of RAB4A, RAB5A, and RAB11A expression levels as specific predictors of pCR to indicated treatment. Results are organized by the logistic model/data used along the rows, and the biomarker evaluated along the columns. Circle sizes are proportional to the significance (−log10 (LR test $p$)); and circle color reflects the magnitude of coefficient (red: positive, blue: negative) from each corresponding logistic model. White background indicates $p < 0.05$, and the odds ratio associated with 1 standard deviation increase in expression were also shown (in white) inside the circle. The 95% confidence intervals for the coefficients are found in Supplementary Table 1. Source data are provided as a Source Data file.

Although we cannot directly compare the two experimental arms, analysis of the trastuzumab + paclitaxel + pertuzumab group ($n = 44$) showed no significant correlation between RAB5A and pCR (Coef = 0.38 (95% CI: −0.36–1.17), LRp = 0.32) suggesting pertuzumab is not the key contributor to the significant pCR and

RAB5A association for T-DM1 + pertuzumab-treated patients (Fig. 4B).

RAB5A is a constitutively expressed protein, and utilization as a biomarker will depend on the establishment of a threshold that separates RAB5A$^{high}$ and RAB5A$^{low}$ expressing cancer. Figure 5A

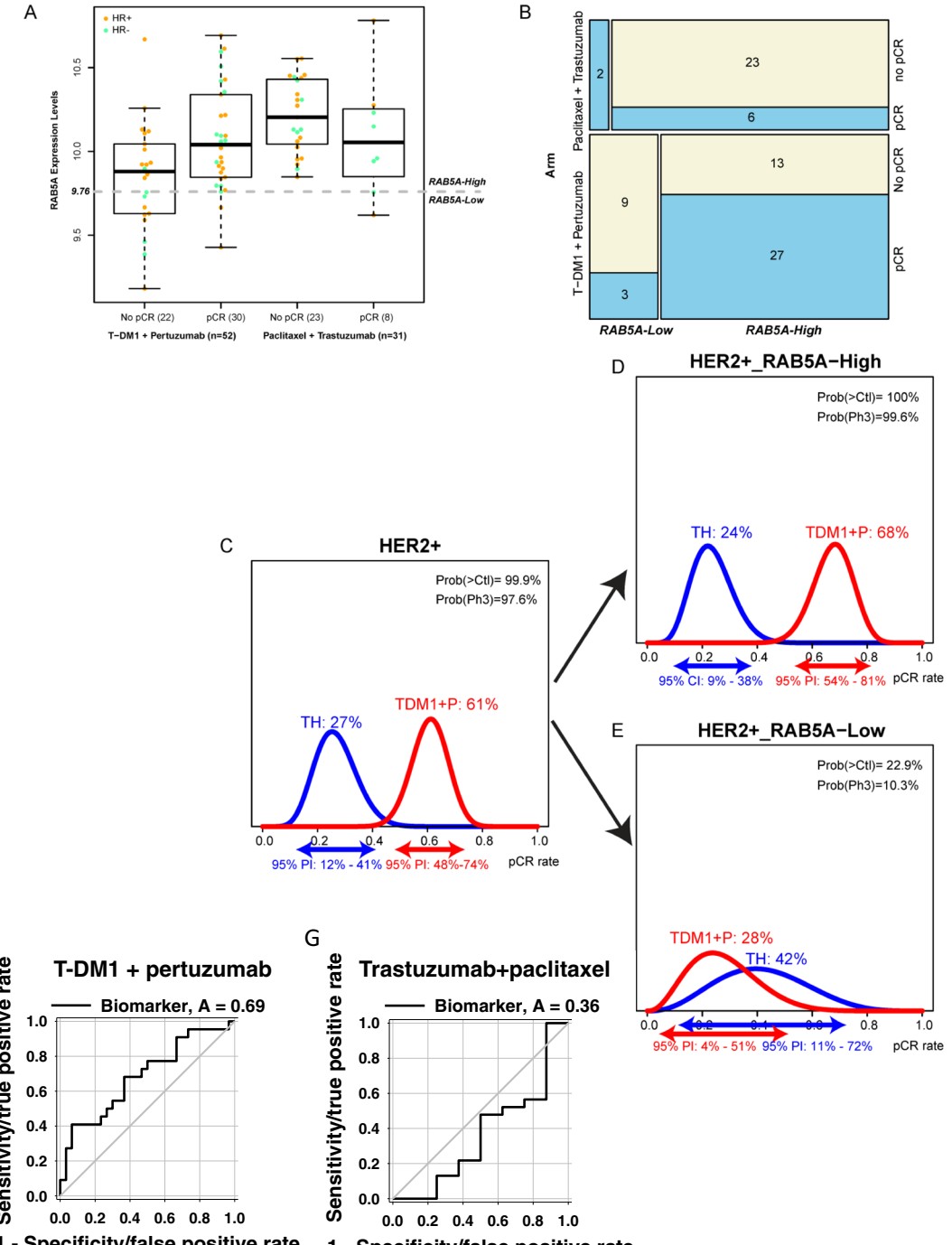

**Fig. 5 Patient distribution within pCR and RAB5A expression in I-SPY2. A** Boxplots of RAB5A expression levels stratified by arm and pCR status (*N* = 83). The midline represents the median of RAB5 expression levels within each group (T-DM1 + pertuzumab-treated, no pCR: *N* = 22 independent samples; T-DM1 + pertuzumab-treated, pCR: *N* = 30 independent samples; trastuzumab + paclitaxel-treated, no pCR: *N* = 23 independent samples; trastuzumab + paclitaxel-treated, pCR: *N* = 8 independent samples). The upper and lower limits of the box correspond to the 1st and 3rd quartile of RAB5A expression, respectively, with whiskers extending to 1.5 times the interquartile range from top/bottom of the box. Dots represent expression values for each individual; and color reflects subtype (orange: HR+; green: HR−). **B** Mosaic plot showing patient distribution within the RAB5A RNA-high and -low based on the threshold of 9.76 by arm and pCR status. **C–E** The bayesian estimated pCR rates within the two treatment groups overall (**C**) as well as when divided in RAB5A RNA-high (**D**) and low (**E**) subsets. **F**, **G** ROC curves of the performance of RAB5A RNA as a biomarker in T-DM1 + pertuzumab-treated (**F**)- and trastuzumab + paclitaxel-treated (**G**) patients. Source data are provided as a Source Data file.

shows boxplots of RAB5A expression levels stratified by arm and pCR status; based on a Monte-Carlo 2-fold cross-validation procedure as described in M&M, a normalized RAB5A RNA expression level of 9.76 was selected as a threshold (dotted gray line). The patient stratification into RAB5A^high and RAB5A^low

groups in each arm is illustrated in Fig. 5B together with the pCR data.

Using this optimal threshold, only 2 patients in the TH arm (6%) had RAB5A levels <9.76 and are considered RAB5A^low, as opposed to 23% (12/52) of the T-DM1 + P arm (Fig. 5B). This

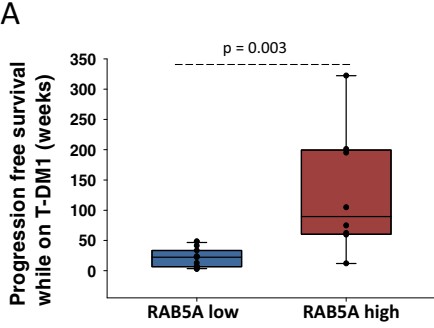

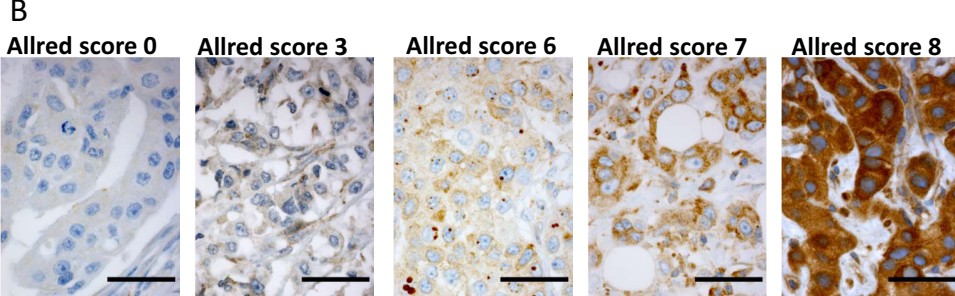

**Fig. 6 T-DM1 and RAB5A correlation in a subset of patients in KAMILLA. A** Box blot indicating the distribution of PFS in RAB5A low (Allred score 0–6, $N = 11$ independent samples) and RAB5A-high (Allred score 7 and 8, $N = 8$ independent samples) expressing patients. The midline represents the median of progression-free survival. The upper and lower limits of the box correspond to the 75% and 25% percentile of progression-free survival respectively; the whiskers represent the 90% and 10% percentile. The data points represent the individual data points in each group. A Mann–Whitney rank-sum test was used to determine the statistical difference in the median values between the two groups. **B** Exemplified images (×40 objective) of RAB5A IHC for Allred score 0 (patient #20), Allred score 3 (patient # 19), Allred score 6 (patient #21), Allred score 7 (patient #24) and Allred score 8 (patient #15). Scale bars: 100 μm. All samples were manually screened by 2 observers. Consensus on staining intensity and amount was obtained, and representative areas defined. Number of images captured from each slide: patient #20 4 images, patient #19 10 images, patient #21 8 images, patient #24 6 images, patient #15 4 images. Source data are provided as a Source Data file.

may in part be attributed to the difference in RAB5A expression between the two arms, where the RAB5A levels are significantly higher in the trastuzumab+paclitaxel arm than the T-DM1 + pertuzumab arm. Nevertheless, within the T-DM1 + P arm, 68% (27/40) of RAB5A[high] patients achieved a pCR, in contrast to 25% (3/12) of RAB5A[low] patients (Fig. 5B). Of note, no significant differences in HR status distribution or pre-treatment ERBB2 expression levels are observed among the RAB5A[high] and RAB5A[low] T-DM1 + P treated patients (% HR−: 30% vs. 42%, Fisher exact test two-sided $p = 0.49$, and median ERBB2: 11.2 (SD: 0.85) vs. 10.6 (SD: 1.27), Wilcoxon rank-sum two-sided $p = 0.08$, respectively).

Bayesian logistic modeling was used to estimate the pCR probability distributions within the T-DM1 + P and TH control arms in the overall HER2[+] population as well as within the (predicted sensitive) RAB5A[high] and (predicted insensitive) RAB5A[low] subsets (Fig. 5C–E). The Bayesian estimated pCR probability is 68% (95% PI: 54–81%) in the T-DM1 + pertuzumab arm relative to 24% (95% PI: 9–38%) in the control arm in the RAB5A[high] patients. In contrast, the estimated pCR probability is 28% (95% PI: 4–51%) in the RAB5A[low] subset in the T-DM1 + pertuzumab arm and 42% (95% PI: 11–72%) in the trastuzumab + paclitaxel arm. For comparison, using the same model, the estimated pCR probability of the entire HER2[+] group is 61% (95% PI: 48–74%) in the T-DM1 + pertuzumab arm and 27% (95% PI: 12–41%) in the trastuzumab + paclitaxel arm.

Finally, we also established ROC curves to find the most appropriate cut-off for RAB5A expression in the T-DM1 + pertuzumab arm. Using this method, the optimal RAB5A cutoff was found at 9.76, the same as the one identified using the Monte-Carlo procedure. The AUC was 0.69, again indicating

RAB5A as a predictive biomarker in this cohort (Fig. 5F). A ROC curve was also calculated for the TH control arm. (Fig. 5G). In contrast to the T-DM1 + P arm, the low AUC of 0.36 (below 0.5) suggested low RAB5A expression as a potential predictor for trastuzumab + paclitaxel response, although the number of patients ($n = 2$) with RAB5A[low] in this arm is too small to evaluate (Fig. 5G).

**RAB5A protein expression correlates with T-DM1 sensitivity in a subset of patients in the Kamilla study.** Our positive correlation between RAB5A expression and T-DM1 sensitivity both in vitro and in the I-SPY2 clinical cohort was further verified in a small subset of the KAMILLA study including 19 patients treated at Oslo University hospital with biobanked primary biopsies. A highly significant ($p < 0.03$) difference in PFS was found between patients with RAB5A Allred scores 7 and 8, where 7/8 patients had a PFS of 50 weeks or more, and Allred scores 0-6, where all patients had PFS less than 50 weeks (Fig. 6A). The Allred score and PFS data for each patient is included in the data source file and exemplified IHC images are shown in Fig. 6B.

## Discussion

Most of the targeting drugs currently approved for the treatment of cancer are mAbs or small-molecular inhibitors for which the drug target also represents the target for the mechanism of action. The target itself represents a clear biomarker for treatment with these drugs, although other factors may be important to identify patients likely to experience resistance or low tolerability. In complex targeting therapeutics incorporating a cytotoxic component such as in ADCs, other biomarkers associated with the

intracellular transport and/or cytotoxic mechanism of action are likely to impact the therapeutic outcome[12,13]. In this study, the need for separate predictive biomarkers for ADC and mAbs treatment is emphasized by the lack of coherence between trastuzumab and T-DM1 sensitivity in the selected panel of HER2-positive cell lines. We demonstrate a linear correlation between cellular HER2 expression and response toward T-DM1 (Fig. 2C). This is in agreement with several clinical studies demonstrating higher response rates of T-DM1 in patients with HER2 mRNA levels above the median compared to the below-median subgroup[14,15,20]. Importantly, despite its highly targeted mechanism, T-DM1 shows clinical benefit only in a subset of HER2- positive breast cancer patients with an objective response rate reported to ~40%[8,21,22]. Furthermore, in HER2-positive gastric cancer, an objective T-DM1 response rate of only ~20% with no increase in efficacy compared to taxanes is reported[23]. The diverse clinical response toward T-DM1 in HER2-expressing cancer indicates drug efficacy to depend also on other factors than the extracellular target expression.

T-DM1 is dependent on endocytosis in order to transport its cytotoxic payload (emtansine) into the cell and we here present the first report on T-DM1 efficacy correlated to proteins involved in endocytosis and endocytic trafficking. Of the three proteins investigated (RAB4A, RAB5A, and RAB11A), only RAB5 expression was found to correlate with T-DM1 toxicity in a cell line panel of HER2-positive breast and ovarian cancer. The correlation of T-DM1 sensitivity and RAB5A protein expression was stronger than observed for HER2 expression, and the highest correlation was found when combining RAB5A and HER2 protein expression indicating incorporation of RAB5A together with HER2 as a better predictive biomarker for T-DM1 sensitivity. RAB5A is localized to early endosomes and regulates both endocytosis and endosome fusion of clathrin-coated vesicles[24]. The lack of correlation observed for RAB4A and RAB11A may reflect the intracellular processing of T-DM1 upon uptake. RAB4A-mediated recycling from early endosomes may, e.g., be limited[25–27] and it is possible that DM1 escape the endocytic vesicles prior to accumulation in RAB11A-positive recycling endosomes. We confirmed the correlation between RAB5A expression and sensitivity to neoadjuvant T-DM1 in the I-SPY study. Even though the number of patients is small, our clinical data from I-SPY2 also illustrates the possibility to define a RAB5A expression threshold to determine T-DM1 treatment in HER2-expressing breast cancer patients. A subset of patients from the Kamilla study was here used as a verification cohort. As compared to the I-SPY2 cohort, the patients from the Kamilla cohort all had advanced progressive disease and had all previously been exposed to different treatment regimens, including chemo- and HER2-targeted therapies. Nevertheless, a highly significant correlation was found between RAB5A expression in primary tumor and PFS while on T-DM1 therapy. Altogether, our presented in vitro findings showing a significant correlation between T-DM1 treatment outcome and RAB5A expression, is here confirmed in two independent clinical cohorts.

It has previously been shown that RAB5A expression predicts poor prognosis of breast cancer patients[28], and the current study strongly indicates that T-DM1 should be further evaluated as the treatment of choice for these patients.

T-DM1 is currently approved only for advanced HER2-positive breast cancer previously treated with trastuzumab, but a defined RAB5A threshold-biomarker holds promise also for patient stratification at earlier stages of this disease[29], in addition to other indications, such as gastric cancer[23].

Although the number of patients with RAB5A[low] receiving trastuzumab in the I-SPY2 cohort is far too low to draw any conclusions, our results may also point toward a negative correlation between RAB5A expression and trastuzumab response. Thus, HER2-expressing cancers with low RAB5A do better on trastuzumab as compared to T-DM1. If low RAB5A can be used not only to deselect patients for T-DM1 but also to find those patients most likely to benefit from trastuzumab, it would be an improvement for future personalized HER2-positive breast cancer therapy. Altogether, better stratification of HER2-expressing breast cancer patients into treatment groups may add to the benefit of the treatment, and the deselected patients may also be offered alternative treatment at an earlier time point.

As the mechanism of T-DM1 action involves endocytosis, we hypothesized that proteins involved in the endocytic process had an impact on the treatment response to T-DM1. Our hypothesis was here confirmed first in vitro and then in two independent clinical cohorts. Furthermore, we believe our results also demonstrate a more general concept in which proteins involved in endocytosis and/or endocytic trafficking are utilized as biomarkers for ADCs. Even though RAB5A was the only candidate to succeed as a predictive biomarker for T-DM1 in the present study, this may be different for other ADCs dependent on both their targeting moiety and cytotoxic payload. Overall, our results imply that current and future ADC development and treatment will benefit from the incorporation of biomarkers reflecting uptake and intracellular transport.

## Methods

**Cells and culturing**. Five HER2-expressing human cell lines were used in this study; the breast cancer cell lines SK-BR-3 (HTB-30), AU-565 (CRL-2351), HCC1954 (CRL-2338), and MDA-MB-453 (HTB-131), and the ovarian cancer cell line SKOV-3 (HTB-77). The human breast cancer cell line MDA-MB-231 (HTB-26) was used as a negative control for HER2 expression. All cell lines were obtained from American Type Culture Collection (ACC)(Manassas, VA, USA), except SK-BR-3, kindly provided by the Department of Biochemistry at Institute for Cancer Research, Norwegian Radium Hospital (the cell lines was originally obtained from ATCC). All cell lines were used between passage number 3 and 25 to avoid changes in the cell line characteristics with time, and the cells were routinely checked for Mycoplasma infections. SK-BR-3 and SKOV-3 cells were cultured in McCoy's 5A medium, AU-565, HCC1954, and MDA-MB-231 cells in RPMI-1640 medium (both obtained from Sigma-Aldrich, St. Louis, MO, USA), while MDA-MB-453 were cultured in Leibovitz's L-15 medium (Lonza, Verviers, Belgium). All media were supplemented with 10% fetal calf serum (ThermoFisher (Life Technologies), Rockford, IL, USA), 100 U/ml penicillin, and 100 µg/ml streptomycin (both from Sigma-Aldrich).

**Cytotoxicity experiments**. Cells were seeded at $8 \times 10^3$ (SK-BR-3), $1.8 \times 10^3$ (SKOV-3), $6 \times 10^3$ (AU-565), $4 \times 10^3$ (HCC1954), or $1 \times 10^4$ cells/well (MDA-MB-453) in 96-well plates (Nunc, Roskilde, Denmark) and allowed to attach overnight. The cells were then incubated with trastuzumab (Herceptin®, Roche, Basel, Switzerland) or T-DM1 (ado-trastuzumab emtansine, Kadcyla®, Genentech, San Francisco, CA, USA) at increasing concentrations for 72 h, after which cell viability was assessed by the MTT assay as previously described. Briefly, cells were incubated with 0.25 mg/ml MTT (Sigma-Aldrich) for 2–4 h before the media was removed and the formazan crystals dissolved in DMSO. Absorbance was measured at 570 nm using a plate reader (PowerWave XS2 Microplate Spectrophotometer, Biotek, Winooski, VT, USA9) and Gen5 software version 2.09 (Biotek). IC$_{50}$ values were calculated from sigmoidal curves (fit model: $a/(1 + \exp(-(x-x0)/b)))$ generated in SigmaPlot extended graph analysis 14 (Systat Software, Inc, Jan Jose, Ca, USA).

**Western blot analysis**. Total cell extracts were obtained from ~80% confluent cells seeded in 6-well plates. The cells were washed once with PBS and collected in 700 µl PBS with a cell scrape on ice before they were subjected to centrifugation at $1000 \times g$ at 4 °C for 5 min. The supernatant was removed and the pellet was kept at −80 °C until lysis. The cell pellets were lysed with 50–150 µl lysis buffer (150 mM NaCl, 50 mM Tris pH 7.5, 0.1% SDS) including Halt protease and phosphatase inhibitor cocktail (ThermoFisher Scientific) on ice for 15–30 min. The lysates were then sonicated and spun down at $12.000 \times g$ at 4 °C for 15 min. The protein concentration in the supernatants was assessed by the Bio-Rad Protein Assay Dye Reagent Concentrate (Bio-Rad Laboratories, Ca USA) before the lysate was transferred to new tubes and stored at −80 °C until SDS-PAGE and western blotting using a Trans-Blot Turbo transfer system (Bio-Rad, Hercules, CA, USA). Cellular protein expression was detected using HER2 (#2165) antibody (1:5000 dilution) from Cell Signaling Technology (Danvers, MA, USA), RAB5A (PA5-29022) (1:2000 dilution), RAB11A (71-5300) (1:1000 dilution), and RAB4A (PA3-

912) (1:1000) antibodies from ThermoFisher Scientific (Rockford, IL, USA). Protein expression was correlated to γ-tubulin as detected by an antibody (#T6557) (1:5000) from Sigma-Aldrich. HRP-linked α-rabbit (#7074) (1:2500) and α-mouse (#7076) (1:2000) antibodies from Cell Signaling Technology were used as secondary antibodies. Supersignal West Dura Extended duration Substrate (Thermo Scientific) and ChemiDoc™ densitometer (Bio-Rad) were used for the detection of protein bands on the membrane. ImageLab 4.1 (Bio-Rad) (software) was used for the quantification of protein expression. The expression of each protein was calculated relative to the highest expressing cell line. Uncropped and unprocessed scans of blots are provided as source data.

**In vitro correlation analysis**. The relative expression of HER2, RAB4A, RAB5A, and RAB11A in the cell lines was plotted against the cell line sensitivity toward T-DM1, as measured by $1/IC_{50}$ (concentration inhibiting the viability by 50%) and a linear regression including the $R^2$ value was assessed using SigmaPlot extended graph analysis 14.

**I-SPY2 TRIAL**. The I-SPY2 TRIAL is a multicenter, open-label adaptive neoadjuvant platform trial for women with breast cancer (>2.5 cm clinically and >2 cm by imaging) including biomarker assessments (based on HER2 status, estrogen and progesterone receptors, and a 70-gene assay (MammaPrint, Agendia)) prior to inclusion. Core biopsy samples are secured for RNA expression analyses, and one of the study aims of I-SPY2 is to test and validate biomarkers for new drugs[30] (NCT01042379). T-DM1 + pertuzumab (T-DM1 + P) were one of the novel combinations evaluated for efficacy in I-SPY2, against a trastuzumab + paclitaxel (TH) control[31]. In the present study, RAB5A, RAB4A, and RAB11A expression were evaluated as specific predictors of pathologic complete response (pCR) to T-DM1 + pertuzumab. Although trastuzumab + paclitaxel + pertuzumab was also evaluated as an experimental regimen over the same period, formal comparisons between two experimental arms are contractually prohibited. Therefore, a separate qualifying biomarker analysis was performed on patients receiving trastuzumab + paclitaxel + pertuzumab to evaluate whether these biomarkers were associated with response to pertuzumab.

*Expression data*. Distinct pre-treatment samples from individual I-SPY2 patients were analyzed on one of two Agilent custom arrays (the 15,746 and 32,627 designs). All samples in the T-DM1 + pertuzumab and trastuzumab + paclitaxel + pertuzumab arm were assayed on the 32627 arrays, while the trastuzumab + paclitaxel arm was split between the platforms, with 22 samples on the older 15,746 platform and 9 samples on the 32,627 array. To combine data across the two designs, the probe annotation of the 15,746 platform was updated (September 2016); and for each platform, collapsed normalized expression data by averaging such that genes represented by multiple probes are computed as the average across probes. The ComBat algorithm was then applied to adjust for platform-biases and combine the data from the two platforms. This procedure was performed for the pre-treatment data of the first 880 I-SPY2 patients irrespective of the experimental arm. Normalized, platform-corrected pre-treatment expression levels of RAB5A, RAB4A, and RAB11A from patients in the T-DM1 + pertuzumab arm ($n = 52$), the trastuzumab + paclitaxel control ($n = 31$), and the trastuzumab + paclitaxel + pertuzumab ($n = 44$) (Source data provided) were evaluated individually for association with pCR without adjusting for multiple hypothesis testing (as described below).

*Qualifying biomarker analysis*. All I-SPY qualifying biomarker analyses follow similar multi-step pre-specified analysis plans[32–34]. Associations with pCR were first assessed within each arm with logistic modeling and significance assessment using the likelihood ratio test (as a one-tailed test of the likelihood ratio statistic against a chi-square distribution with 1 degree of freedom) (lmtest R package v.0.9-37). As well, the interaction between biomarker and treatment was evaluated using a logistic model fitted to data from the T-DM1 + pertuzumab and control arms. These analyses were also performed adjusting for HR status as a covariate.

In a second step, an optimal dichotomizing threshold was determined for biomarkers that specifically associate with response in the T-DM1 + pertuzumab but not the control arm and have a significant ($p < 0.05$) biomarker × treatment interaction, using a Monte-Carlo 2-fold cross-validation procedure. Specifically, for 100 iterations, half of the cases were randomly selected, balancing for treatment arm and pCR status, as training set. Every value between the 10th and 90th percentile were considered as a potential threshold to dichotomize the training set into "High" vs. "Low" RAB5A expressing groups; and fit a series of logistic regression models to assess the biomarker × treatment interaction. The threshold which minimizes the likelihood ratio (LR) test $p$-value for the interaction term in the training set was selected and used to dichotomize the test set, and assess the significance of the biomarker x treatment interaction in the test set. The LR $p$-values across the 100 test sets were then combined using the logit method (metap R package v1.4); and the threshold yielding the minimum combined LR test $p$-value was selected.

In the final step, the optimal threshold identified to dichotomize patients into RAB5A high and RAB5A low groups was used to calculate the Bayesian estimated PCR probability in the T-DM1 + pertuzumab and control arms using a Bayesian covariate-adjusted logistic model similar to the standard one I-SPY2 uses to evaluate agent efficacy[35] (but without time-adjustment). Specifically, pCR is modeled as a function of subtypes defined by HR status, HER2 status, and RAB5A (low vs. high group), treatment arm, and the interaction between subtype and treatment. We assumed independent normal prior distributions N(0,1) for each of the model coefficients; and Markov chain Monte-Carlo sampling was performed using rjags (Martyn Plummer 2019. rjags: Bayesian Graphical Models using MCMC. R package v4-10) (10,000 iterations with 1000 iterations for adaptation, and a burn-in of 5000). No new codes were generated for this study. Commands and packages in R were used to do all the statistical analysis.

*Establishment of receiver operating characteristic (ROC) curves*. The patients in each treatment group were sorted according to RAB5A expression. ROC curves were established to visualize how the expression level correlated with pCR. The SigmaPlot extended graph analysis 14 software was used for the establishment and analysis of ROC curves. The T-DM1 + P curve was also used to identify a threshold for RAB5A expression to dichotomize patients into RAB5A-high and -low.

**KAMILLA trial**. The KAMILLA trial (NCT01702571) is an international multicenter single-arm, open-label phase IIIb safety study of T-DM1 including patients with HER2-positive advanced breast cancer with progression after prior treatment with chemotherapy and a HER2-directed agent for metastatic disease[36]. The Norwegian Radium Hospital, OUS was included as one of the centers recruiting patients in the Kamilla study, and primary FFPE biopsies were available for 19 of the patients. The patients received T-DM1 until unacceptable toxicity, withdrawal, or disease progression. The data from the patients included in the current study were retrieved directly from the medical records of the patients. Three patients, all with a progression-free survival of more than 50 weeks, either stopped treatment due to toxicity or were lost to follow-up. All other patients discontinued treatment due to the progression of the disease. In the present study, these primary biopsies were subjected to IHC using an anti-RAB5A from Abcam (ab 109534; rabbit IgG, clone EPR5438, diluted 1:1600; incubation 30 min on a Dako Autostainer platform). Antigen retrieval was performed using Dako's High pH solution in a PT link, and as detection system was used Dako's EnVision Flex+ system with rabbit linker (K8009). Slides from FFPE blocks from SK-BR-3 and SKOV-3 cells were included in the staining runs as positive and negative controls, respectively. Images were captured using a Zeiss Axiophot microscope with a Plan-Neofluar 40x objective (Zeiss, Oberkochen, Germany) and a Leica DFC320 camera (Leica, Wetzlar, Germany). Corel PaintShop Pro x8. Version 18.00.124 (Corel, Ottawa, Canada) was used to process the images together with Image J (1.53C) (LOCI, University of Wisconsin, USA) for insertion of the scale bar. The RAB5A staining, localized to the cytoplasm and/or membrane of the tumor cells, was quantified using the Allred Scoring system, where the staining is evaluated based on both color intensity (0—Negative, 1—Weak, 2—Intermediate, 3—Strong) and proportion of stained cells (0—No positive, 1—≤1% positive, 2—1–10% positive, 3—11–33% positive, 4—34–66% positive, 5—67–100% positive)[37,38]. RAB5A expression, as measured by Allred score, was correlated to progression-free survival (PFS) data in a dot plot. A cut-off of Allred score 7–8 was selected based on this plot showing all patients with progression-free survival >50 weeks to score 7 or 8, and all except one patient with progression-free survival < 50 weeks to score 0-6 in the Allred system. All measurements were taken from distinct samples.

**Ethics oversight and consent**. Both the I-SPY2 and Kamilla study complied with all relevant ethical regulations for clinical trials with human participants. The I-SPY2 study was approved by IRB boards at the participating study sites:
    *Institution: University of California San Diego Moores Cancer Center
    Name and address IRB: University of California, San Diego Human Research Protections Program Institutional Review Boards (Attn: Human Research Protections Program (HRPP) Altman Clinical and Translational Institute, Level 2 9452 Medical Center Drive La Jolla, CA 92093).
    *Institution: Georgetown University Lombardi Cancer Center
    Name and address: MedStar Health Research Institute-Georgetown University Oncology Institutional Review Board (Medical-Dental Building, SW104, 3900 Reservoir Road NW, Washington, DC 20057).
    *Institution: Loyola University Chicago Stritch School of Medicine, Cardinal Bernardin Cancer Center
    Name and address: Loyola University Chicago Health Sciences Division Institutional Review Board for the Protection of Human Subjects (2160 South First Avenue Maywood, IL 60153).
    *Institution: University of California, San Francisco, Helen Diller Family of Comprehensive Cancer Center
    Name and address: UCSF Human Research Protection Program Institutional Review Board (490 Illinois Street, Floor 6, San Francisco, CA 94143).
    *Institution: University of Texas, Southwestern Medical Center Simmons Comprehensive Cancer Center
    Name and address: UT Southwestern IRB (5323 Harry Hines Blvd. Dallas, TX 75390).
    *Institution: H. Lee Moffitt Cancer Center and Research Institute
    Name and address: Chesapeake IRB (3181 SW Sam Jackson Park Road - L106RI Portland, OR 97239-3098).

*Institution: Oregon Health and Science University Knight Cancer Institute
Name and address: Oregon Health & Science University Research Integrity Office IRB (3181 SW Sam Jackson Park Road - L106RI, Portland, OR 97239-3098).

*Institution: Mayo Clinic Breast Cancer Center - Rochester
Name and address: Mayo Clinic Institutional Review Boards (201 Building, Room 4-60, 200 First St. SW, Rochester, MN 55905).

*Institution: University of Pennsylvania, Abramson Cancer Center
Name and address: University of Pennsylvania Office of Regulatory Affairs Institutional Review Board (3624 Market St., Suite 301S, Philadelphia, PA 19104-6006).

*Institution: University of Alabama at Birmingham Comprehensive Cancer Center
Name and address: The University of Alabama at Birmingham Office of the Institutional Review Board for Human Use (470 Administration Building, 701 20th Street South, Birmingham, AL 35294-0104).

*Institution: University of Minnesota, Masonic Cancer Center
Name and address: University of Minnesota Human Research Protection Program (MMC 820 420 Delaware St. SE, Minneapolis, MN 55455-0392).

*Institution: University of Colorado Cancer Center
Name and address: Colorado Multiple Institutional Review Board (COMIRB) (University of Colorado, Anschutz Medical Campus, 13001 E. 17th Place, Building 500, Room N3214, Aurora, CO 80045).

*Institution: University of Washington Medical Center
Fred Hutchinson Cancer Research Center (FHCRC) IRB (Institutional Review Office 1100 Fairview Ave. N. Mail Stop J2-100, Seattle, WA 98109).

*Institution: University of Southern California, Norris Comprehensive Cancer Center
University of Southern California Health Sciences Institutional Review Board (LAC + USC Medical Center, General Hospital Suite 4700, 1200 North State Street, Los Angeles, CA 90033).

*Institution: University of Texas, M.D. Anderson Cancer Center
University of Texas MD Anderson Cancer Ctr Clinical IRBs (Office of Human Subjects Protection Unit 1637, 7007 Bertner Ave., Houston, TX 77030-3907).

*Institution: Swedish Cancer Institute
Western Institutional Review Board (WIRB) (1019 39th Avenue SE Suite 120, Puyallup, WA 98374-2115).

*Institution: University of Arizona, Arizona Cancer Center at UMC and UMC-North
University of Arizona Institutional Review Board (The University of Chicago Biological Sciences Division/University of Chicago Medical Center, 5751S. Woodlawn Ave., 2nd floor, Chicago, IL 60637).

All patients recruited in I-SPY2 signed an informed consent form.

The use of the Kamilla samples in the present study, was approved by the institutional research review board (Oslo University Hospital, Department of Cancer, Po Box 4953 Nydalen, 0424 Oslo), and the Regional Committees for Medical and Health Research Ethics (REC North-Secretariat, University of Tromsø, Po Box 6050 Langnes, 9037 Tromsø). All patients enrolled in the Kamilla study provided written informed consent. The patients still alive at the time of collection of data presented here, signed an additional informed consent.

**Reporting summary**. Further information on research design is available in the Nature Research Reporting Summary linked to this article.

## Data availability

All raw data necessary to interpret, verify, and extend the research in this article is provided in the data source files. Source data are provided with this paper.

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

## Acknowledgements

We thank the South-Eastern Norway Regional Health Authority for financial support and Dr. Sebastian Patzke for help processing the IHC images.

## Author contributions

O.E. was responsible for gathering, processing, and analyzing the Kamilla data and was responsible for the outline of the study together with K.B. and A.W. E.B. and Ø.G. established and conducted the Rab5A quantitative IHC. C.Y., A.M., L.v.V., and L.E. are all included in the I-SPY2 team, C.Y. was responsible for the statistical evaluation of the I-SPY2 data, A.M. is the chaperone of the relevant arms of the I-SPY2 study used here and is also chair of the I-SPY2 clinical trial operations group. L.v.V. is the chair of the I-SPY2 biomarker working group. L.E. is PI of the I-SPY2 trial. M.E.B.B., A.S.V.F., and A.W. were responsible for generating the in vitro data. K.B. was responsible for processing and analyzing the in vitro data. A.W. was the project leader and was involved in the evaluation of all the research data, as well as the main responsible for the manuscript. All authors discussed the results and commented on the manuscript.

## Competing interests

A patent application entitled "Diagnosis and treatment of cancer" by Anette Weyergang, Kristian Berg, Olav Engebraaten, and Maria E.B. Berstad, application number: WO 2018/234872 A1 is currently in the national phase.
