## [Peer Review File · Nature Communications]

Reviewers' Comments:

Reviewer #2:

Remarks to the Author:

Interesting report that offers some insight to possible biomarker for TDM-1 response based on a biologic mechanism. As a short report I think it is reasonable for publication and does offer a potential way of identifying patients suitedor not.....for this therapy. One of the acknowledged limits of this paper is the relatively few patient samples available for validation....ie KAMILLA

Reviewer #3:

Remarks to the Author:

This manuscript describes a sequence of procedures to establish potential biomarker predictive associations with TDM-1 sensitivity using cell lines, the I-SPY2 trial, and validation in the KAMILLA trial. The results of this sequence suggest that RAB5A was the strongest candidate as a predictive marker for TDM-1 efficacy.

There is not much to comment on statistically. The cell line data use a standard approach to estimate IC50 generated using software from SigmaStat. While some of the correlations are quite high, this can happen when there is a low-low point and a high-high point that drive the fitted equation to linearity. Nonetheless, the RAB5A results are convincing.

Statistical methods for the I-SPY2 are prescribed by the trial and include a Bayesian model of logistic regression on the marker with the endpoint of pCR while adjusting for other factors. The methods reasonably show an association of RAB5A with TDM1+pertuzumab versus control.

Finally, the KAMILLA trial was used to provide validation. The only point that lacks clarity is why the cutpoint of 7-8 on the Allred scale was chosen rather than some other cutpoint.

Overall, the analytic methods are strong with the only concern being the method to choose the cutpoint on the Allred score in the analysis of the KAMILLA trial.

Reviewer #1

Interesting report that offers some insight to possible biomarker for TDM-1 response based on a biologic mechanism As a short report I think it is reasonable for publication and does offer a potential way of identifying patients suitedor not.....for this therapy. One of the acknowledged limits of this paper is the relatively few patient samples available for validation....ie KAMILLA

We thank the reviewer for this positive feedback. We agree that the numbers of patients in both the I-SPY2 and Kamilla cohort are low and this is communicated clearly in the manuscript. Nevertheless, we believe the strength in our research findings lies in a biological sensed hypothesis, which was confirmed first in vitro and then in two independent clinical cohorts. We are currently in the process of finding larger patients cohorts where Rab5A can be evaluated as a biomarker for T-DM1.

Reviewer #2

This manuscript describes a sequence of procedures to establish potential biomarker predictive associations with TDM-1 sensitivity using cell lines, the I-SPY2 trial, and validation in the KAMILLA trial. The results of this sequence suggest that RAB5A was the strongest candidate as a predictive marker for TDM-1 efficacy.

There is not much to comment on statistically. The cell line data use a standard approach to estimate IC50 generated using software from SigmaStat. While some of the correlations are quite high, this can happen when there is a low-low point and a high-high point that drive the fitted equation to linearity. Nonetheless, the RAB5A results are convincing.

Statistical methods for the I-SPY2 are prescribed by the trial and include a Bayesian model of logistic regression on the marker with the endpoint of pCR while adjusting for other factors. The methods reasonably show an association of RAB5A with TDM1+pertuzumab versus control. Finally, the KAMILLA trial was used to provide validation. The only point that lacks clarity is why the cutpoint of 7-8 on the Allred scale was chosen rather than some other cutpoint. Overall, the analytic methods are strong with the only concern being the method to choose the cutpoint on the Allred score in the analysis of the KAMILLA trial.

We thank the reviewer for this positive feedback. The cut point of 7-8 on the Allred scale was chosen after plotting PFS (weeks) versus Allred score of all included patients. The cut off point of 7-8 for more than 50 weeks PFS was visualized in this graph showing all patients with progression free survival > 50 weeks to score 7 or 8. Also, all except one patient with PFS < 50 weeks had Allred scores 0-6. We agree that this should be clarified in the M&M section and have corrected paragraph 2.6 accordingly.